

# A Comprehensive Global Mapping of Offshore Lighting

Christopher D. Elvidge[1], Tilottama Ghosh[1], Namrata Chatterjee[1,2], Mikhail Zhizhin[1,3], Paul C. Sutton[2], Morgan Bazilian[4]

[1] Earth Observation Group, Payne Institute for Public Policy, Colorado School of Mines, Golden, Colorado 80401 USA
[2] Department b of Geography, University of Denver, Denver, Colorado 80401 USA
[3] Space Research Institute, Russian Academy of Sciences, Moscow, Russia
[4] Payne Institute for PublicPolicy, Colorado School of Mines, Golden, Colorado 80401 USA

*Correspondence to*: Chris Elvidge (celvidge@mines.edu) Orcid ID 0000 0003 0584 1098

**Abstract.** We present the first comprehensive multiyear global mapping of offshore lighting structures derived from low-light imaging satellite observations collected at night. The sensor is the day/night band (DNB) flown as part of the NASA/NOAA Visible Infrared Imaging Radiometer Suite (VIIRS). The product merges two operational nighttime light products: VIIRS boat detection (VBD) data and VIIRS cloud-free nighttime lights (VNL). The two products are spatially complementary, making it possible to fill gaps through a merger. Both product sets have an average DNB radiance layer, and the merger involves preserving the higher of the two average radiances. A wide range of lighting structures are present, from fishing grounds, platforms, anchorages, gas flares, transit routes, and the glow surrounding bright lighting onshore. The richness in the numbers and types of offshore lighting structures traces back to the DNB spike detector at the core of the VBD algorithm. The VNL algorithm uses outlier removal to filter out biomass burning, an essential process for mapping electric lighting onshore. The outlier removal drops about 80% of the offshore lighting detections. We expect the new product will lead to an improved understanding of fishing grounds, offshore light pollution, and supply chain disruptions at anchorages, thereby aiding in the development of more sustainable and efficient practices.

## 1 Introduction

In 2015-2016 I heard a frequent question from users of the satellite boat detection data my team produced and distributed. The algorithm reports the location and brightness of offshore lights at night from NASA-NOAA Visible Infrared Imaging Spectrometer Suite (VIIRS) day/night band (DNB) data in near-real time (Elvidge et al. 2015). The initial VIIRS Boat Detection (VBD) production area was Indonesia and the Philippines, under contract with the U.S. Agency for International Development. We provided near-real-time VBD data, technical support, and training to the two governments. They used VBD data to monitor fishery closures, marine protected areas, and fishing grounds. Both governments continue using the VBD data to this day.

The most common question from the users was how they could distinguish boats using lights for fishing versus other offshore lighting. My response was, "If a VBD pixel is in a fishing ground – it is likely fishing". It turned out that Indonesia





and Philippines divide their waters up into regional sections, and neither had a specific "fishing-ground map." With no existing map – we decided to make our own with temporal compositing of VBD data. The temporal compositing worked – making it possible to delineate fishing grounds and label lights associated with fishing grounds. In addition to fishing grounds, several

other types of features emerge – piquing our interest. VBD processing was expanded to global in 2017, clearing the way for a multiyear VBD product.

Vessel detection is one of the core activities of maritime domain awareness, used in near-real-time surveillance and building long-term chronological records of offshore human activities. There are three basic styles of satellite detection of vessels: literal, track, and electromagnetic emission detection. Literal vessel detection relies on reflectance from the physical

structure of vessels (Kanjir et al. 2018). For example, daytime detection of reflective pixels embedded in a dark sea surface in the near or shortwave infrared images collected by optical imagers, such as Landsat (McDonnell and Lewis, 1978; Wu et al., 2009; Gellert 2023). Synthetic Aperture Radar is another data source widely used in literal vessel detection (Vachon et al., 2000). Vessel track phenomena detectable from space include wakes (Liu and Deng, 2018; Graziano et al., 2016) and condensation trails (Ferek et al., 1998). Electronic emissions from vessels provide a third style of boat detection from space.

This includes locational beacons such as the Automatic Identification System (AIS) (Tetreault, 2005) and regionally available Vessel Monitoring System (VMS) (Shepperson et al. 2018). AIS is required for vessels over 300 gross tons (GT) worldwide, while VMS requirements only exist for specific countries and typically start at 30 GT or less. Some vessels lacking AIS or VMS are detectable by radio frequency (RF) emissions recorded by specialized sensor constellations such as Hawkeye 360 (Hawkeye 2020). Several teams have used these data sources to generate global maps of offshore human activity (Watson et

al., 2004,2013,2018; Gelchu and Pauly, 2007; Halpern et al., 2008, 2015, 2019; Anticamara et al., 2011; Kroodsma et al., 2018; Andrello et al., 2022; Paulo et al., 2024).

Another type of vessel EM emissions detectable from space is the radiant emissions from electric lights. Electric lights are widely deployed on fishing boats, particularly in Asia, to attract catch. This collection capability, first recognized in the 1970s (Croft 1979), comes primarily from polar-orbiting meteorological imagers with panchromatic low-light imaging

spectral bands specifically designed to detect moonlit clouds (Miller et al., 2013). Two sensor series have collected this style of data. The original is the U.S. Air Force Defense Meteorological Satellite Program (DMSP) Operational Linescan System (OLS), which collects nightly global data at coarse spatial resolution (2.7 km ground sample distance) and no radiometric calibration. OLS digital archives extend from 1992 to now, with data from nine sensors. The NASA/NOAA Visible Infrared Imaging Radiometer Suite (VIIRS) day-night band (DNB) is the second sensor series that detects lighting at the Earth's surface.

The VIIRS DNB nightly low-light imaging record extends from 2012 to the present with pixel footprints of 742 x 742 meters and in-flight radiometric calibration to radiance units.

Since the mid-1990s, the Earth Observation Group (EOG) has produced monthly and annual cloud-free composites from DMSP and VIIRS low-light imaging data (Elvidge et al., 2022). The generation of satellite-derived nighttime light data follows a set of filtering and averaging steps, including the following: ( a) Exclusion of sunlit and moonlit data. Sunlight blocks

the detection of light at the Earth's surface. Moonlit data are excluded from the products to prevent the inclusion of non-lighting





features such as moonlit clouds, snow-covered terrain, and high albedo land surface features. (b) Exclusion of clouds, which obscure and blur surface lighting features. (c) Biomass burning is filtered out of the annual nighttime lights through outlier removal. This filtering is essential since biomass burning dominates the detected features in fire-prone areas (Elvidge et al., 2022). (d) The averaging includes all cloud-free observations meeting the other filtering criteria. This averaging style results

in the dimming of ephemeral lights, such as vessel detections. The results are annual grids, free of biomass burning, showing the locations and top-of-atmosphere average brightness levels for electric lighting on land from 65 south to 75 north latitudes.

The filtering and averaging steps in EOG's annual nighttime lights result in the loss of nearly 80% of the DNB-detectable offshore lighting. The loss is highest for fishing grounds and less for fixed location persistent offshore lighting features, such as platforms. Despite the heavy loss of offshore lighting features, EOG's annual nighttime lights form the basis

for two global maps of offshore light pollution (Davies et al., 2014 and Smyth et al., 2021).

In addition, two published studies of offshore fishing grounds based on EOG's monthly nighttime light products lack outlier removal. The monthly products have a more complete rendition of offshore lighting, but the radiance averaging includes cloud-free pixels lacking lighting, reducing the apparent brightness of lit boats. In one case, the authors (Zhao et al., 2018) applied a land-sea mask to 36 months of EOG's global VIIRS cloud-free average radiance images, spanning 2014-2016. This

product includes many fishing grounds, gas flares, and narrow strips of dim lighting offshore from coastal cities. Having 36 consecutive months of data made it possible to generate temporal profiles, and the authors reported that most fishing grounds had distinct annual cycling, which was absent at gas flaring and shoreline glow areas. The authors provide several colorized maps of features found, but the original digital product is inaccessible. The other paper (Li et al., 2021) used seven years of EOG monthly VNL to study fishing activities in the South China Sea.

In 2015 EOG developed a nightly VIIRS boat detection (VBD) data product optimized for offshore vessel detections (Elvidge et al., 2015). VBD uses a DNB spike detector to identify offshore lighting and two additional indices to rate the amplitude and sharpness of the spikes.   VBD runs over clouds and does not record the cloud state for detection. Specialized modules adaptively raise the detection threshold to reduce false detection from moonlit clouds and specular reflectance of moonlight in lunar glint zones. EOG supplies nightly VBD to fishery agencies and other organizations, primarily in Asian

countries. EOG also builds VBD monthly and annual summary grids, which differ from the VNL products in five aspects: (a) VBD temporal summary grids only cover offshore areas. (b) Average radiances only include pixels having DNB spike detections, with no dilution of the average from overpasses lacking detection. (c) With nightly VBD detections, it is possible to calculate the percent frequency of offshore lighting detection. (d) The summary grid products include data from all lunar conditions. And (e) VBD reports detections made through optically thin clouds.

This paper reports on a new comprehensive mapping of offshore lighting spanning multiple years, primarily based on EOG's nightly VIIRS boat detection (VBD) product. The global offshore lighting grids from 2012 to 2022 are available for download via links listed in Elvidge *et al.,* 2024. In addition, three papers have reported preliminary regional results from the VBD multiyear composite (Elvidge et al., 2023a and b; Chatterjee et al., 2023).




## 2    Methods

### 2.1 VIIRS boat detection data

In 2015, the Earth Observation Group developed the VIIRS boat detection (VBD) algorithm with support from NOAA's Joint Polar Satellite System (JPSS) proving ground program and the U.S. Agency for International Development
(USAID). The arrival of new DNB granules triggers VBD processing in near real-time, with data distributed to several fishery agencies and other organizations. In Asia, the nightly VBD record extends back to April 2012. The global VBD record begins in 2017. VBD's primary products include nightly csv and kmz files. A database has been built and is used to make global temporal composites and temporal profiles from individual features.

The VBD detections are binned into eleven classes, called Quality Flags (QF), based on the spike sharpness and
location (Table 1). The flare quality flags are defined based on EOG's infrared emitter catalog (Elvidge *et al.*, 2023c). The Earth Observation Group developed the VIIRS boat detection (VBD) algorithm in 2015 with support from NOAA's Joint Polar Satellite System (JPSS) proving ground program and the U.S. Agency for International Development (USAID). VBD operates in near-real time, triggered by the arrival of new data, and distributed to several fishery agencies and other organizations. In Asia, the nightly VBD record extends back to April of 2012. Global VBD record begins in 2017.

When moonlight is either absent or extremely dim (less than or equal to 0.0001 lux), the SMI thresholds are set at a constant level for each detector aggregation zone. These thresholds are set slightly above the noise floor. This results in a stepwise increase in thresholds moving from the nadir to the edge-of-scan. When moonlight is present, these low thresholds result in vast numbers of false detections from moonlit clouds. To address this problem, EOG developed a method for adaptively raising the SMI detection threshold to minimize the number of false detections. This
adaptive thresholding is performed with the cross-correlation of the SMI versus the longwave infrared radiances from imagery band 5 (I5). A separate VBD module identifies zones where lunar glint is possible and adaptively sets detection thresholds to exclude DNB spikes associated with specular reflectance from the sea surface. The VBD detections are binned into eleven classes, referred to as Quality Flags (QF), based on the spike sharpness and location (Table 1). The quality flags for flares are defined based on EOG's infrared emitter catalog (Elvidge et al., 2023c).





**Table 1**

135                 VBD Detection Types

| Quality Flag | Number | Composite |
|---|---|---|
| Strong | 1 | X |
| Weak | 2 | X |
| Blurry | 3 | |
| Flare | 4 | X |
| High Energy Particles | 5 | |
| Lunar Glint | 6 | |
| Glow | 7 | X |
| Recurring | 8 | X |
| Sensor Crosstalk | 9 | X |
| Weak and blurry | 10 | X |
| Platform | 11 | X |

### 2.2 Multiyear VBD Composite

140       Four types of multiyear (2012-2022) 15-arc-second composite grids were generated from the VBD database: coverages, detection tally, average radiance, and percent frequency of detection. The detection tally and average radiance included VBD detections from quality flags 1, 2, 4, 7, 8, 10, and 11. The set includes three sets of detection types having lower amplitude DNB spikes, including weak, glow plus weak and blurry. Their selection is based on their spatial features matching VBD open ocean density boundaries present in multiyear composites of the QF1 strong

detections. For East and Southeast Asia, the multiyear composite has VBD data from 2012-2022. Outside of Asia, the composite has VBD data from 2017 to 2022. VBD follows the sparse style of grid filling (Figure 1), where only the grid cell with the detection pixel center location is filled in (Elvidge et al., 2023c).





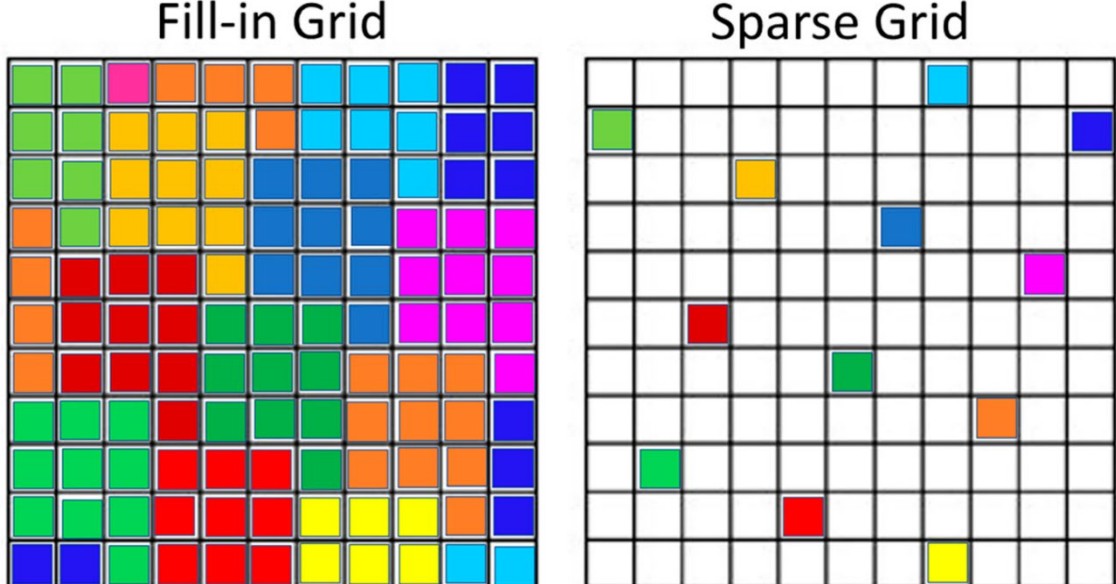

**Figure 1.** Fill-in versus sparse geolocation for VIIRS pixels from a portion of a single sub-orbit into
a 15 arc second grid. The colors represent the pixel identities. In sparse gridding, only the 15 arc second grid cells
containing the latitude/longitude of the VIIRS pixel center is filled in. The "fill-in" style of geolocation starts from the
sparse grid, followed by nearest-neighbour resampling to ensure all the grid cells are filled. From Elvidge *et al.,* 2023c.

**2.3 Multiyear VNL Composite**

155        A second 15 arc second VIIRS nighttime lights composite was generated by merging eleven years (2012-2022)
of VNL of cloud-free nighttime lights average DNB radiance grids with background masked to zeroes. The methods
used to make each of the annual VNLs are standardized (Elvidge *et al.*, 2022). The eleven years of VNL were downloaded
https://eogdata.mines.edu/products/vnl/#annual_v2. A land-sea mask is applied to the multiyear average and used
as one of the depictions of offshore lighting. VNL follows the fill-in-gridding approach shown in Figure 1.


**2.4 Comparison and potential merger of VBD and VNL composites**

The offshore areas and feature content of the VBD and VNL composites are compared. The two sets of grids differ from
each other (Table 2), but both sets contain average radiance. We evaluate whether combining the two based on average
radiance would create a more comprehensive global map.






**TABLE 2**

175                          VBD versus VNL Summary Grids

| GRID TYPE | VBD | VNL |
|---|---|---|
| Coverages | X | X |
| Cloud-free Coverages | | X |
| Detection Tally | X | |
| Percent Frequency of Detection | X | |
| Average Radiance | X | X |

## 2.5 Temporal Profiles

We built a capability to extract nightly and monthly VBD temporal profiles based on vector polygons for specific cluster
features, including fishing grounds, platforms, gas flares, and anchorages.

## 2.6 National Offshore Lighting Index

The percentage of offshore areas having lighting detected from 2012-2022 was calculated for each country by dividing
the lit surface area by the total offshore area calculated for that country's exclusive economic zone (EEZ).

## 3    Results

### 3.1 Comparison of VBD and VNL Lighting Areas

190        Four styles of binary masks record the spatial extents of detections in VBD, VNL offshore, cells with both VBD and
VNL, and grid cells where VBD average radiance is greater than or equal to the VNL average radiance. The surface area
under each binary mask is area-corrected to compensate for the reductions in 15-arc second surface areas as a function
of latitude. This makes it possible to report the area as square kilometers in Table 3. VBD covers four times more areas
than offshore VNL. The two products are meticulously merged (or blended) by taking the grid cell with the largest
average radiance, ensuring the most accurate representation of the data extent. The merged total records the maximum
extent of detected offshore lighting at 15-arc second resolution. The percentage of merged total with VNL only is
8.6%. The rate of the merged area with VBD only is 79.6%. The percentage of the merged area having detection in both



VBD and VNL is 11.7%. Where both VBD and VNL are present, VBD's average radiance is greater than the VNL radiance for 98% of the area. The total area in the merged product covers 1.4% of the offshore total.


**TABLE 3**

VBD and VNL Surface Areas

| Category | Area (km$^2$) |
|---|---|
| Offshore Total | 343,359,529 |
| VBD | 4,220,715 |
| VNL Offshore | 940,030 |
| Both | 542,594 |
| Unique VBD | 3,678,121 |
| Unique VNL | 397,436 |
| Merged total | 4,618,151 |
| VBD .ge. VNL where both present | 536,068 |
| % of merged area with only VNL | 8.61% |
| % of merged area with only VBD | 79.64% |
| % of merged area with both | 11.75% |
| % VBD .ge. VNL where both are present | 98.80% |
| Merged total % of offshore | 1.34% |
| % of merged total lacking VNL | 79.64% |

**3.2 Side-by-side Comparisons of VBD, VNL, and merged set**

On a global scale, VBD detections cover four times more area than offshore VNLs. This section examines side-by-side examples of the two product sets as grayscale images and a color composite showing the blending. Figure 2 has VBD and VNL subsets for a fishing ground in the Arafura Sea in Indonesia. Land areas are marked dark green. The
upper pair comes from VBD with the percent detection frequency on the left and average DNB radiance in the upper right. The most significant fishing ground extends south from Aru Island and has an elliptical cloud of detections that wraps around the island's southern tip. There is an empty buffer between the fishing ground and the shoreline, an expression of compliance with the government's ban on commercial fishing within 12 nautical miles of shore (Government of Indonesia, 2023).    The VNL set has the original multiyear average DNB radiance in the low left with
shoreline vectors. VNL has very few zones of onshore lighting in this part of the remote Arafura Sea. The masked VNL average DNB radiance is in the lower right. The VNL fishing ground features cover less area and have a blurry appearance compared to VNL. Fewer fishing ground detections allow for the averaging process of all-night, cloud-free observation. The sharpness appears blurry compared to VBD due to the differing methods used to fill the 15-arc second





grid. VBD takes a sparse grid approach, while VNL takes a fill-in grid approach (Figure 1). The merged average
radiance grid was sourced almost entirely from VBD, which has higher average DNB radiances than the corresponding
multiyear VNL (Figure 3).

The second example presents an area with a relatively even mix of VBD and VNL offshore lighting features - a
portion of the Persian Gulf surrounding Qatar. Figure 4 provides a clear comparison, with the VBD set on top and the
VNL set below. The VBD set includes the percent detection frequency in the upper left and the average radiance on the
upper right. The VNL average DNB radiance image in the lower left corner has not been masked to zero out onshore
lighting, making it easier to associate offshore glow with bright lighting present onshore. Notably, lighting from gas
flares in the Persian Gulf forms circular features in both VBD and VNL, with radiance tapering towards the outer edges.
This practical example underscores the versatility of both VBD and VNL in different geographical settings.

Figure 5 is a color composite that reveals the source of the VBD / VNL merged product. The merged product
reports the average radiance for the product having the larger average DNB radiance. The average radiance coming
from VNL is red, and the average radiance from VBD is cyan. In the case of shorelines with glow, such as along the
shorelines of Qatar, Bahrain, and the United Arab Emirates, VBD was selected over VNL. This decision was based on the
specific characteristics of the glow and the detection capabilities of VBD. VBD is also chosen in the gaps between VBD
detections in the glow surrounding flares, further demonstrating the strategic use of VBD in remote sensing data
analysis.

Key differences between the offshore VBD and VNL multiyear grids in the Persian Gulf include: (a) VBD has
less glow detected than VNL surrounding bright lighting onshore, for instance, along the eastern shore of Qatar. (b)
VBD detected glow surrounding gas flares have a fragmented appearance with many gaps between grid cells having
VBD detection. (c) VBD missed the glow surrounding one of the large flares. And (d) Many gas flares have a near
vertical (north-south) high average radiance stripe. For (a) and (b), the paucity of VBD detections offshore from major
cities and surrounding gas flares traces back to the spike detector used in VBD and the land mask bump-out that
occurred mid-way through the VBD series production. For (c)- the missing flare glow is the result the flare being on a
small island, which resulted in a land mask VBD bump out that thoroughly covered the flare glow. For (d), a form of
hysteresis called crosstalk causes a vertical alignment of bright VBD detections to either side of a large gas flare. Gas
flares are notable as the brightest objects recorded by the DNB. DNB crosstalk is a false detection style found at
extremely bright point sources (typically gas flares) aligned perpendicular to the scan direction.



**Figure 2.** Comparison of VBD and VNL products for a fishing ground in the Arafura Sea, Indonesia. The VBD set includes

the percent frequency of detection and DNB average radiance. The VNL set consists of the original average DNB and land-masked average radiance.



**Figure 3.** Combined VBD and VNL images of the Arafura Sea show where VNL average radiance exceeded VBD in red and where VBD exceeded VNL radiance in cyan.



# Comparing VBD and VNL - Persian Gulf

VBD  Percent Frequency of Detection          VBD  Average DNB Radiance

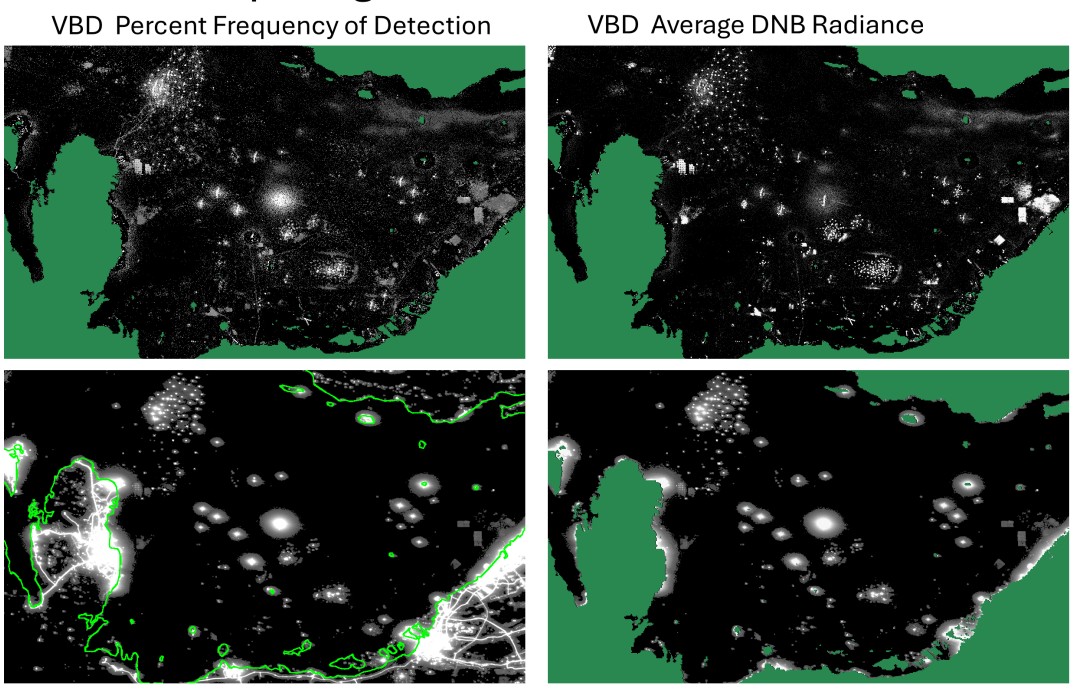

VNL Average DNB Radiance          VNL Average DNB Radiance Sea Masked

**Figure 4.** Comparison of VBD and VNL products for a fishing ground in a portion of the Persian Gulf surrounding Qatar. The VBD set includes the percent frequency of detection and DNB average radiance.  The VNL set consists of the original average DNB and land-masked average radiance.



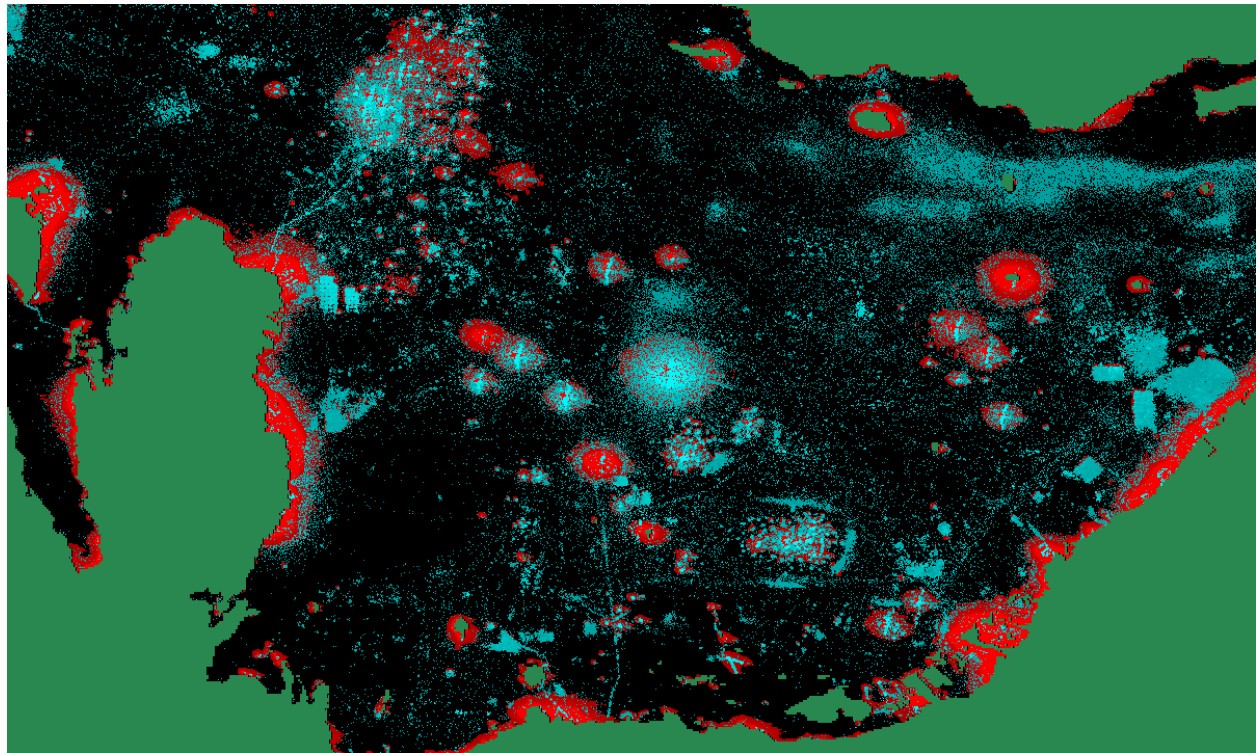

**Figure 5.** Combined VBD and VNL images of the Persian Gulf show where VNL average radiance exceeded VBD in red and where VBD exceeded VNL radiance in cyan.

### 3.3 Offshore Lighting Structure Types

Primary features identified in the VBD multiyear grids include diffused fishing grounds, recurring detections, lit platforms, gas flares, anchorages, and transit lanes. The lit platforms and gas flares are fixed location facilities frequently evident in the monthly VBD composites. Fishing grounds and anchorage structures are incomplete in individual monthly composites, and their full extent may take multiple years to emerge. In this section, we examine features in the multiyear VBD detection grid using sample images from several parts of the world.

#### 3.3.1 Diffused Fishing Grounds

Fishing ground features range from amorphous clusters of VBD detections to boundary-constrained clusters. Figure 6 shows an amorphous fishing ground in the Arafura Sea. An empty buffer surrounding the main island is associated with the government's ban on commercial fishing within 12 nautical miles of shore. There is a sharp vertical line, with a higher percentage frequency of detection on one side – located on the southwest side of the main island. This may be related to the relaxation of the 12 nm boundary at some point in the temporal record. There is a





second fishing ground to the northeast of Aru Island. This fishing ground consists of a loose cluster of detections having a 'salt-and-pepper" appearance. Similar examples of diffused fishing grounds have been identified in many parts of the world. The amorphous outline of fishing grounds suggests that populations of catch species define the spatial extents.


### 3.3.2 Recurring Detection Sites

Individual 15-arc second grid cells within amorphous fishing grounds typically have low percentage detection frequencies – generally under 1%. In shallow waters, there is often a random scattering of grid cells with 2 to 5% detection frequencies referred to as "recurring detections." Figure 6 shows an example of recurring detections from the Arafura Sea. There is something special about those grid cells, which boosts their percent detection frequency compared to their surroundings. However, the exact causes for the detection frequency boost have not been determined. Possible options include fish-aggregating devices (FADS) locations on buoys, lift-net platforms, or floating service centers.


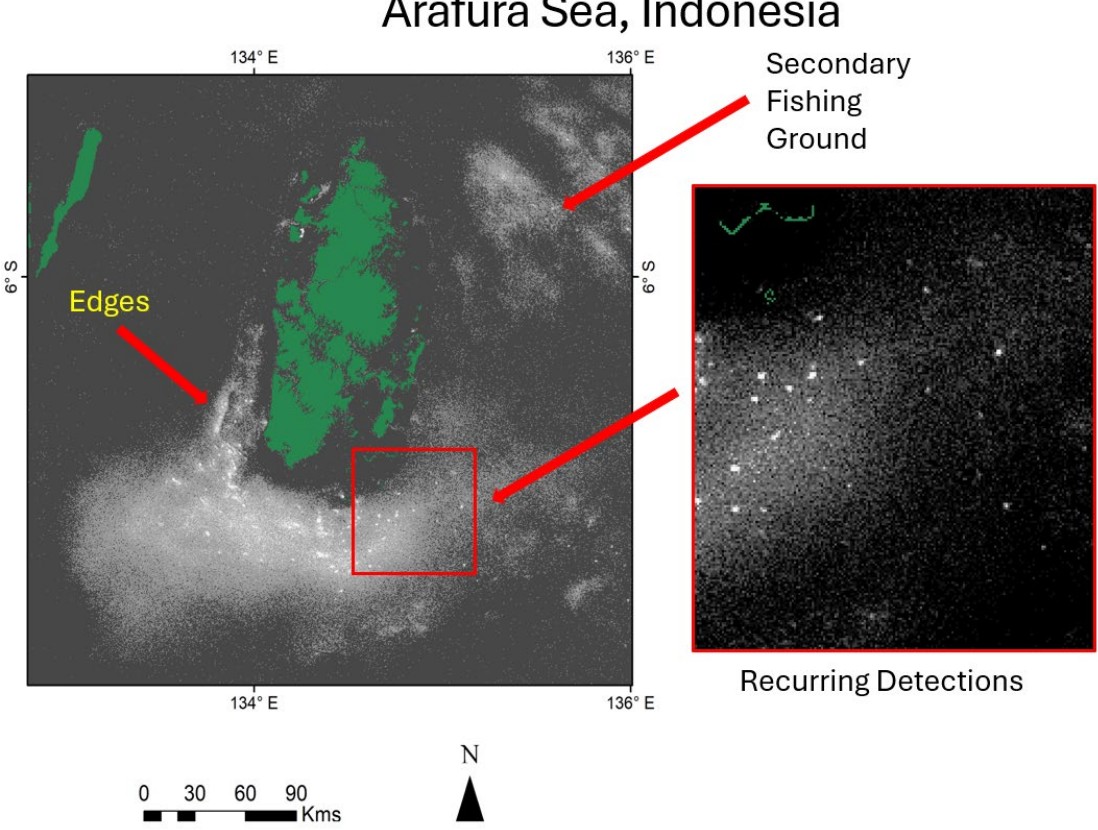

**Figure 6.** VBD percent detection frequency for fishing grounds in the Arafura Sea, Indonesia. The more considerable fishing ground is south of Aru Island – with a diffused "salt-and-pepper" pattern with detection density tapering at the edges. In addition, conspicuous grid cells have more significant percent detection frequencies – referred to as recurring detection sites.

### 3.3.3 Grids and Linear Recurring Detections

This style of fishing ground is recognized based on grids or linear features having regularly spaced "anchor" grid cells with substantially higher percentage frequency of detection. The individual anchor points are similar to the recurring detections found in amorphous fishing grounds regarding appearance and percentage detection frequency. However, the anchor cells are on evenly spaced grids and lines. The best examples of grid fishing grounds are in the Thailand EEZ, especially in the Gulf of Thailand (Figure 7). The Thailand side of the gulf has an unusual regular grid pattern, with one nautical mile spacing and aligned north-south and east-west. The anchor grid cells have many more detections than their surroundings. The Figure 7 inset image in the lower left shows the grid pattern near the island of Koh Tao. The island's government banned commercial fishing near shore, and the width of this buffer zone was



expanded midway through the VIIRS record, resulting in a double ring in the VBD detection numbers. The second inset image shows the junction between Vietnam, Thailand, and Malaysia. The Vietnam waters show a salt-and-pepper" detection pattern with no indication of grids or linear clusters. On the Thailand side, the grid pattern typical to the

north begins to transition to linear features near the Malaysian EEZ line. On the Thailand side, the lines of dense detection grid cells lack the cardinal direction alignment found to the north.   On the Malaysia side, east-west oriented strings of evenly spaced grid cells with high VBD detections.

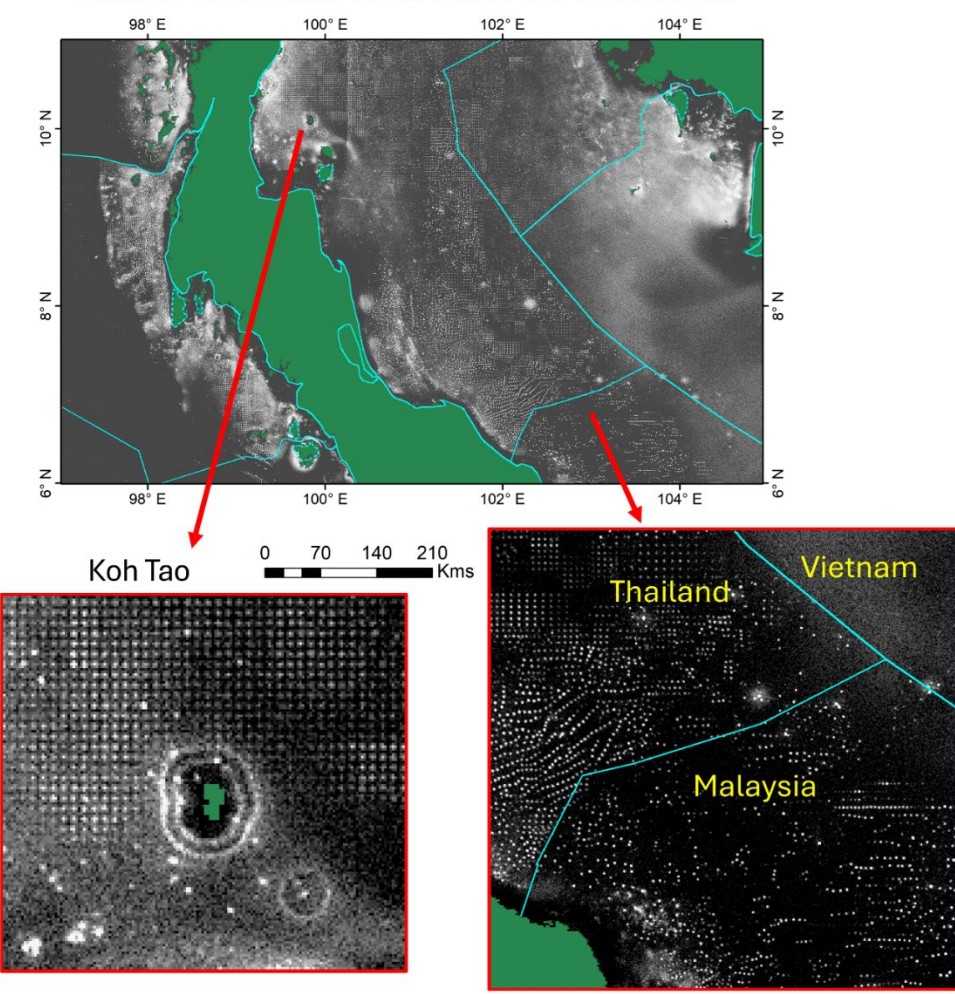

**Figure 7.** VBD percent detection frequency for portions of the Andaman Sea and Gulf of Thailand. Insets show the grid
pattern common across much of Thailand's offshore fisheries. The center grid cells have 20 to 50 times more detections than the adjacent cells. In Thailand, many grids align to the cardinal directions, and the spacing between grid centers





is one nautical mile. Approaching the Malaysian EEZ line, the grids morph into diagonals. On the Malaysian side, there are east-west lines of equally spaced anchor grid cells with high numbers of VBD detections.

### 3.3.4 Fishing Grounds with Artificial Boundaries


There are several fishing grounds where the density of fishing boat activity changes abruptly along a line, indicating an artificial linear boundary. Some of these are EEZ (Exclusive Economic and Zone) boundaries, but others indicate boundaries established in bilateral agreements granting a country's fishing fleet authorization to fish in another country's waters (Amin et al., 2022). Several examples are present in the seas surrounding Japan, Korea, and
China. Figure 8 has a sharp set of boundaries not aligned with EEZ lines. The lines are inside the South Korean EEZ and the Japan-Korea Joint Use Area. Chinese fishing boats have authorization to fish up to the vertical lines in the middle of the zones. The density of fishing boats is higher on the west side of the lines where Chinese fishing boats are permitted. The density of VBD detection is highest along the line's edge as fishing boats use lights to attract catch from over the line.





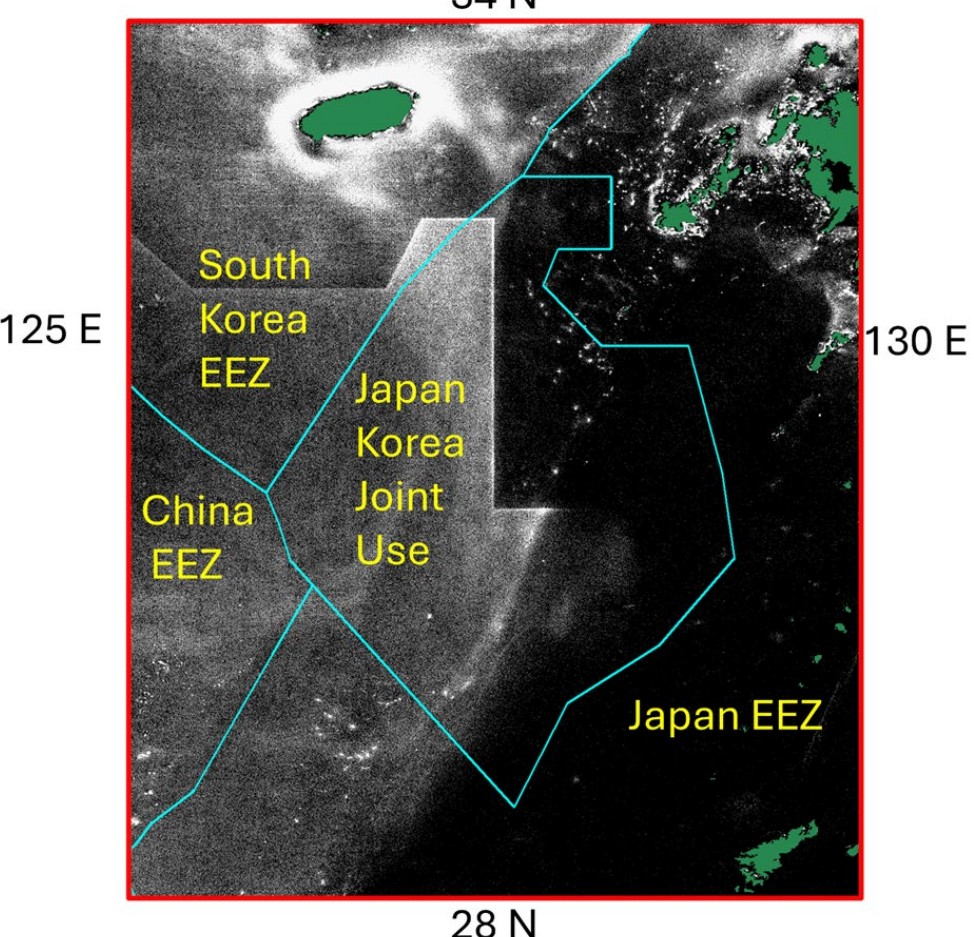

**Figure 8.** Multiyear VBD percent frequency of detection southwest of Japan. The EEZ boundaries are in cyan and labeled in yellow. Distinct linear features cross the South Korean EEZ and Japan-Korea Joint Use areas. These boundaries are where the percent frequency of detection changes along a sharp boundary not aligned with the EEZ lines. These trace back to bilateral agreements granting Chinese fishing boats authorization to fish inside other EEZ zones.




### 3.3.5    Fishing Grounds with Convoluted Boundaries

Several fishing grounds have distinctly convoluted boundaries. One example of this is offshore from Palawan Islands in the southwest Philippines (Figure 9).

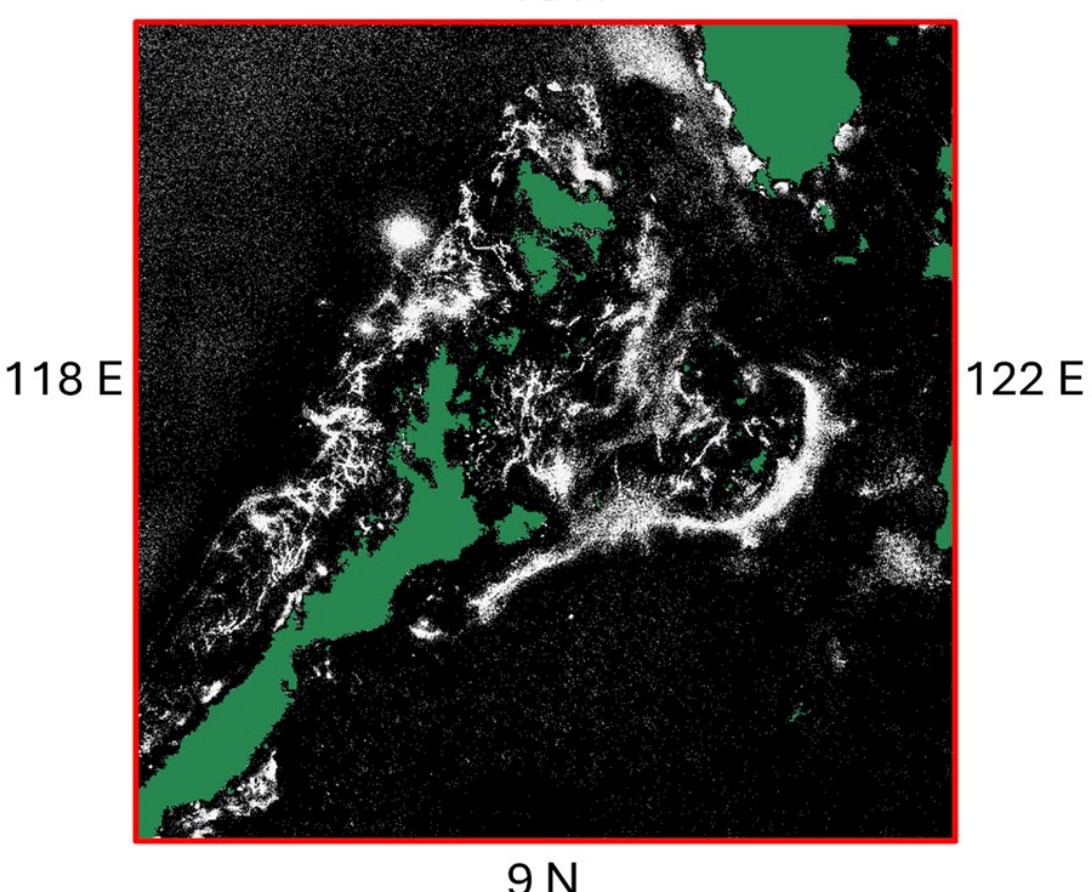


**Figure 9.** Multiyear VBD percent detection frequency surrounding the northern end of the Palawan Island in the Philippines. The fishing grounds have sharp, convoluted edges.



### 3.3.6 Anchorages

These are not just compact clusters of VBD detections, but unique areas with linear edges or rectangular outlines found near ports and narrow sea passages with heavy vessel traffic. Vessels wait in anchorages to enter port or transit a sea strait. Anchorages stand out based on their higher percentage VBD detection frequency than their surroundings. Anchorages near Singapore are in Figure 10, and examples in the Bohai Sea (China) are in Figure 11. The largest anchorages near Singapore are to the east and have sharp boundaries and several low VBD transit lanes, a

distinct feature of this area. Smaller bright patches on the Malaysia side of the Straits of Malacca are likely anchorages, another unique characteristic. There are rectangular and linear fishing grounds east of Peninsular Malaysia, a pattern not commonly seen in other areas. Further east, there are bright spots in Indonesia's Natuna Sea, perhaps from lift net platforms using lights to attract catch, a unique fishing method. The Bohai Sea area of China (Figure 11) has many offshore lighting structures, including many anchorages, platforms, flares, and linear transportation tracks, a unique

combination.  The Bohai Sea has some of the largest anchorages found in the world, a significant fact.



**Figure 10.** Multiyear VBD percent frequency of detection surrounding Singapore. The area has multiple anchorages east of Singapore and several smaller anchorages on the Malaysian side of the Straits of Malacca. Fishing grounds are present to the east of Peninsular Malaysia.



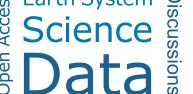

**Figure 11.** Multiyear VBD percent frequency of detection in the Bohai Sea of China. The area has multiple anchorages, platforms, and transit corridors.



### 3.3.7    Transit Lanes

The best examples of VBD transit lanes are passenger ferry tracks going from one port to another. Figure 11
shows several examples in the Bohai Sea VBD image. The Mediterranean Sea is another area with many passenger ferry tracks. Shipping lanes can have sparse tracks in the multiyear VBD composite, such as the sparse tracks through the Straits of Malacca in Figure 10. Another style of transit track present in the multiyear VBD product are air transit routes, including stacked routes over the North Atlantic Ocean. The air transit routes arise from VBD detections from moonlit aircraft.

### 3.3.8    Cat Eyes

The cat eyes phenomenon, a unique offshore lighting feature, is a fascinating blend of atmospheric scattering of light surrounding large flares and a DNB sensor artifact known as "crosstalk". This intriguing phenomenon has been studied since the mid-1970s when it was discovered that large gas flares have distinct glow haloes in low-light imaging visible band data collected by DMSP satellites at night (Croft, 1979 and Elvidge *et al.*, 2009). Glow haloes surrounding
natural gas flares are also observed in the VIIRS DNB, as seen in the Persian Gulf in Figure 4. In addition to the atmospheric glow surrounding large flares, there is an additional DNB feature called crosstalk. This phenomenon, a hysteresis effect Mills, 2016 and Mills and Miller, 2016), results in signal leaks from extremely bright DNB pixels containing flares to an along-track set of pixels from the same scan. The affected pixels have much higher radiances than the surrounds but lower radiance than the flare pixel. Crosstalk accumulates in the multiyear VBD composite as a
slit-like bright feature surrounded by a circular glow pattern – forming a feature reminiscent of a cat's eye (Figure 12). The multiyear VBD composite excluded the VBD detections label as "crosstalk' in the near-real-time nightly products. The current crosstalk identification is based on a vector polygon set drawn on a limited duration temporal composite from the 2015-16 time range. The nightly labeling of crosstalk is incomplete, indicating the vectors should be redrawn based on the new multiyear composite.



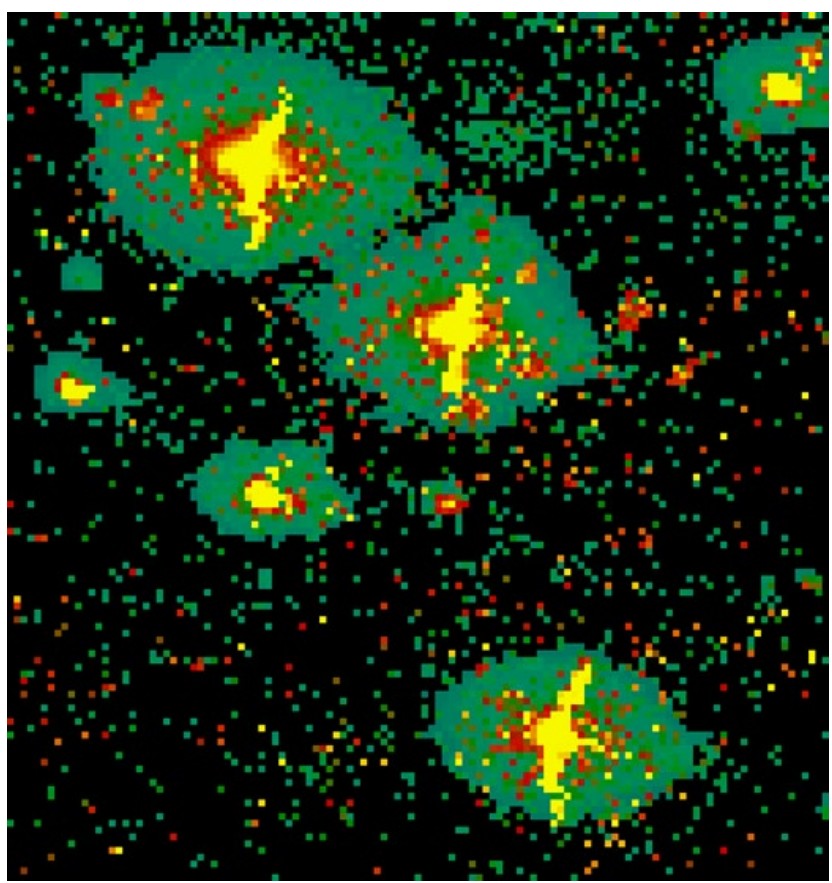

**Figure 12.** Cat eye features are found in the merged VBD and VNL composite in the Persian Gulf (subset from Figure 4). The green features are from the visible glow of scattered light surrounding large gas flares. The DNB crosstalk is a hysteresis effect where radiance leaks to pixels with the same sample number from individual 16-line scans collected by VIIRS.



### 3.4 Temporal Profiles

The individual nightly VBD detections are in a database – making it possible to extract nightly temporal profiles
based on vector polygons established for individual lighting structures. Figure 13-14 have examples of VBD fishing
ground temporal profiles. As noted by other authors, VBD fishing grounds have distinct annual cycling. The Arafura
fishing ground (Figure 13) shows steady increases in detections each year from 2012 to 2019, followed by declines in
2022-2023. The Myanmar fishing ground (Figure 14) also has conspicuous annual cycling and has remained steady
from 2012-2023. The Singapore anchorages (Figure 15) have no yearly cycling and relatively steady nightly detection
numbers from 2012-2020. There are two periods – in 2016 and 2020 – where detection numbers increased to nearly
100 per night. The anchorage detection numbers dropped in 2021 and remained low through 2023, possibly indicating
a supply chain slowdown. The gas flare in the Bohai Sea (Figure 16) shows largely steady activity from 2012 to mid-
2015, followed by five years (2016-2020) of reduced flaring activity and then increased flaring activity in 2022-2023.
The nightly DNB radiance for the flare closely follows the nightly shortwave infrared radiance levels recorded by the
VIIRS Nightfire product (Figure 17). Electric lighting is rarely detected in the shortwave infrared,  implying that the
flare dominates the DNB radiance at the site rather than the electric lighting.

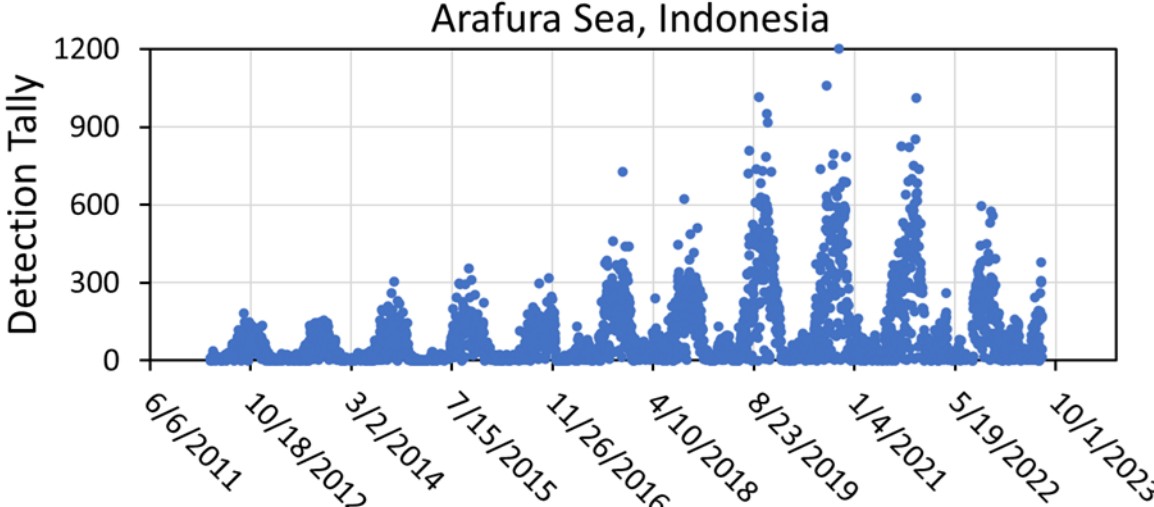

**Figure 13.** The nightly temporal profile of VBD detection tallies for the fishing ground south of Aru Island (Figure 2).
Each year, there is a decrease in detection numbers from January to June, corresponding to the rainy season. The
detection numbers increased consistently from 2012 to 2021, but declined in 2022 and 2023.

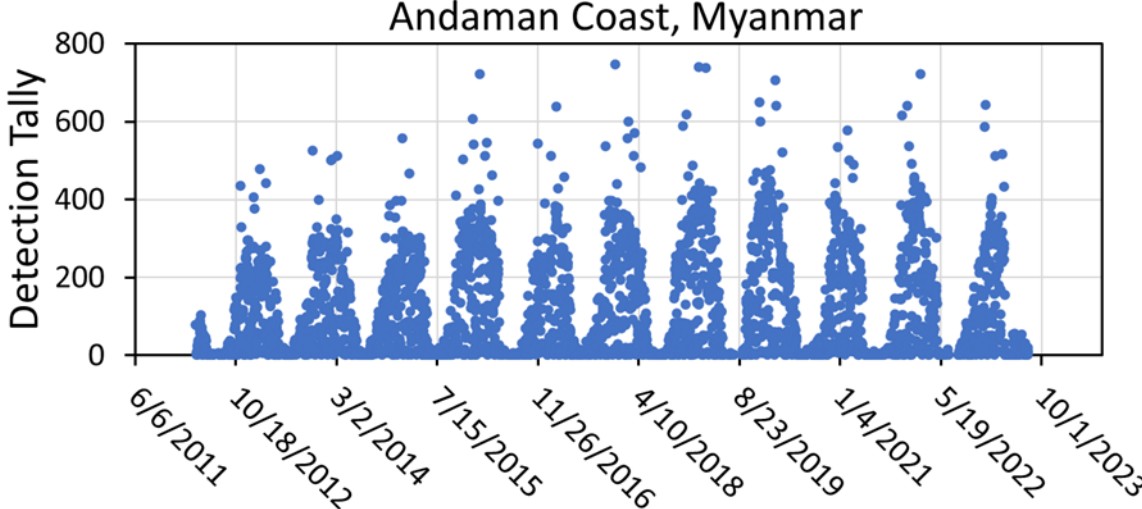

**Figure 14.** VBD detection tally temporal profile for the Myanmar fishery along the Andaman coast. The profile shows vigorous annual cycling, with low detection numbers during the monsoon months (June through September). There is a gradual increase in yearly detections from 2012 through 2020 and then near steady in 2021 through 2022.

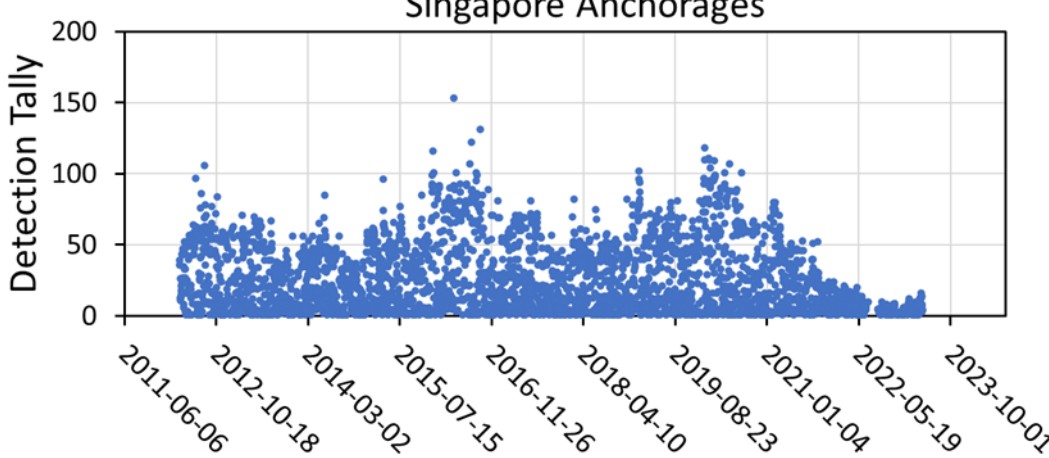

**Figure 15.** VBD detection tally temporal profile for Singapore anchorages, shown in the insert image of Figure 9. The anchorage boat detection tallies ran 50 to 70 per night in 2012 and increased to 100 detections in 2017 and 2020. The detection numbers declined in 2021 and were down to 10-15 per night in 2023.

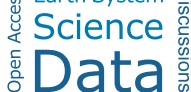

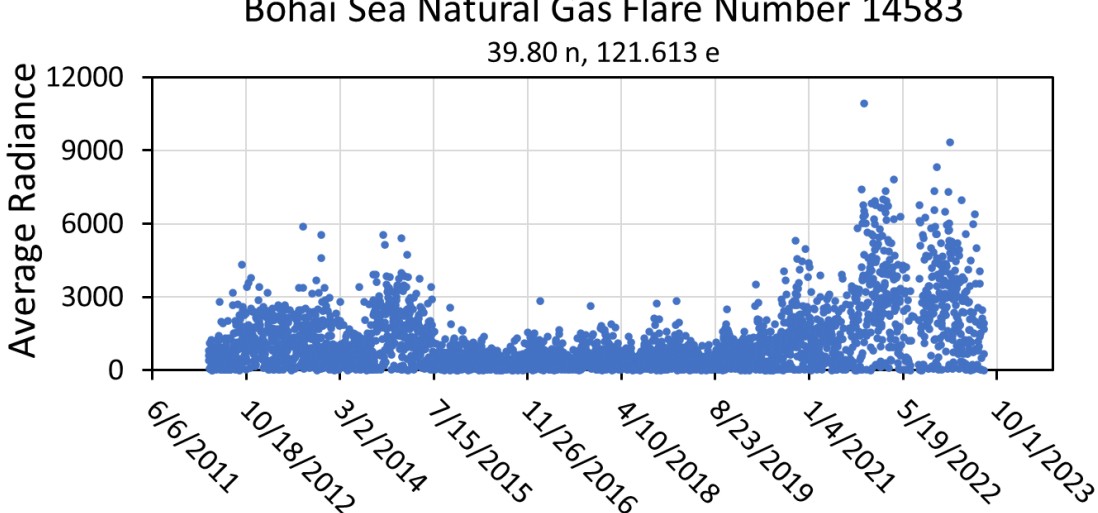

**Figure 16.** VBD radiance temporal profile for a natural gas flare in the Bohai Sea. The radiance starts from 2500 to 3000 from 2012 to mid-2015. Then, the radiance declines to nearly 1000 from 2016 to 2020 before increasing again from 2021. The pattern closely matches the flare's shortwave infrared radiance profile from the VIIR Nightfire time series (Figure 17).


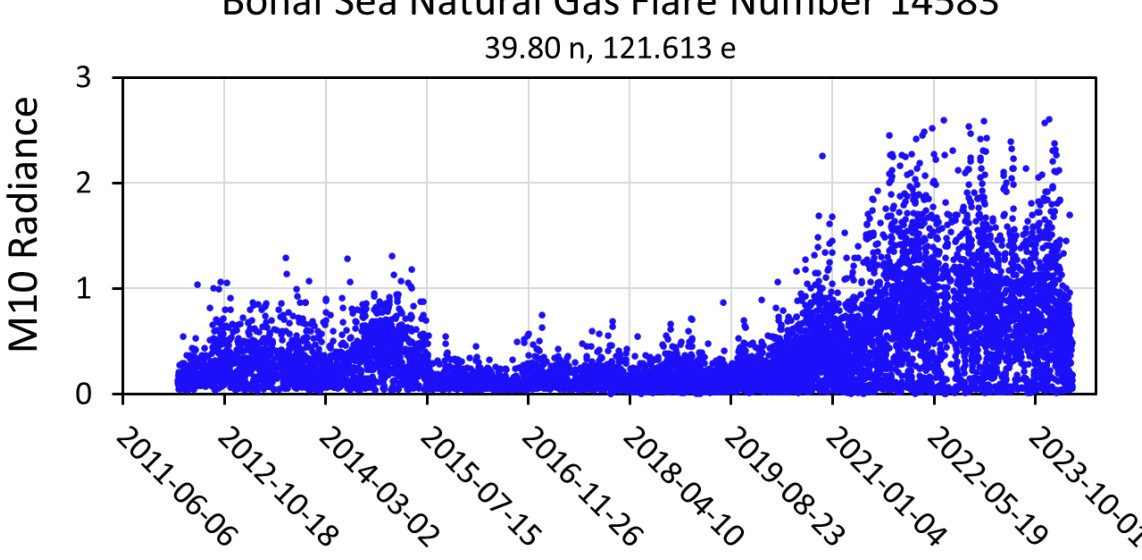

**Figure 17.** Nightly shortwave infrared (band M10) radiance for the Bohai Sea natural gas flare from the VIIRS Nightfire product line. Because the DNB radiances closely track the M10 radiance levels, we conclude that the DNB radiance is dominated by the flare's radiant emissions and not the electric lighting at the site.




### 3.5  National Rankings of the Offshore Lighting Coverages

The spatial extent of offshore lighting in different countries' exclusive economic zones (EEZ) is calculated as a percentage and listed in Table 4. The percentage of offshore lighting is the area of lit grid cells divided by the total EEZ
area. This calculation method allows for an accurate comparison of lighting levels between different countries and regions, as it normalizes for each country's EEZ size. The surface area of each grid cell is adjusted to account for the decreasing size of grid cell areas as latitude increases. The resulting percentages provide valuable insights into the extent of offshore lighting and its potential impact on marine ecosystems. Singapore has the highest percentage, with nearly 100% of its EEZ showing detected lighting. Jordan follows with a small portion of the Gulf of Aqaba, an arm of
the Red Sea. Asian countries, such as Cambodia, Vietnam, South Korea, and Thailand, also show significant spatial extents for offshore lighting. Note that the percentages are based on tallies of lit 15-arc second grid cells and represent areas impacted by artificial lighting. The actual area of lighting (luminaires) is quite small but is readily visible from a considerable distance at sea.

**Table 4**

455                 Ranking of Countries Based on the Percentage of EEZ Having VIIRS Detected Lighting

| Name | Offshore Lit (%) |
|---|---|
| Singapore | 99.87 |
| Jordan | 98.83 |
| Cambodia | 71.88 |
| Vietnam | 61.85 |
| South Korea | 60.53 |
| Thailand | 52.57 |
| Palestine | 51.81 |
| China | 48.30 |
| Iraq | 42.52 |
| North Korea | 40.55 |
| Slovenia | 36.94 |
| United Arab Emirates | 28.75 |
| Belgium | 28.14 |
| Qatar | 25.99 |
| Democratic Republic of the Congo | 24.94 |
| Bahrain | 23.96 |
| Taiwan | 22.84 |
| Malaysia | 22.15 |
| Kuwait | 21.60 |
| Croatia | 17.04 |
| Monaco | 14.69 |
| Myanmar | 14.39 |
| Cameroon | 13.87 |
| Albania | 12.78 |
| Iran | 11.50 |
| Republic of the Congo | 10.90 |



## *4*    **Discussion**

Users should be aware of several critical differences between the VBD and VNL multiyear composites. While both product suites include an average radiance, this is calculated differently and is not directly comparable. VBD calculates its average radiance by summing the radiances of VBD detections in each grid cell and dividing them by the number of detections. It operates without cloud screening and utilizes data from all lunar phases. On the other hand, the VNL's average radiance is derived from the cloud-free DNB radiances from dark portions of lunar cycles, divided by the number of cloud-free observations. The VNL products are filtered annually to remove outliers and background zeroed
out. These distinct characteristics play a significant role in their respective applications.

The VBD multiyear composite stands out by including the percent detection frequency grid, which VNL does not provide. This metric is possible in VBD due to the spike detector at the algorithm's core. While VNL averages cloud-free observations, it does not track the number of times lighting is detected and thus has no calculation of the percent frequency of detection.

The VNL long-term composite excels in depicting the offshore glow surrounding bright sources. However, it falls short in detecting lit fishing boats due to the outlier removal step that filters out biomass burning on land. In contrast, VBD readily detects isolated lit pixels embedded in dark backgrounds, providing a more complete depiction of fishing ground spatial extents and temporal profiles. This clear distinction between their strengths and limitations can guide the readers in choosing the most suitable grid for their specific needs.

Regarding the area results presented in Tables 3 and 4, it is important for the user to recognize that these are based on tallies of 15-arc second grid cells having offshore lighting detected. The actual lighting area is limited to the luminaires and is minuscule compared to the image grid cells. The 15-arc second grid cell tallies represent the ocean surface area affected by artificial lighting on one to many nights during the eleven-year compositing period. We did include a latitude-based area corrected to avoid exaggerating lit area sizes at higher latitudes.








### 5 Conclusion

At night, the ocean's surface is a vast dark palette where the VIIRS day/night band can readily detect faint lighting sources. While EOG produces VBD nightly, the nightly products do not exhibit the sharply defined structures
depicted in Figure 2-11. Fixed locations, such as lit platforms, are able to propagate to the monthly VBD composites. However, more extended periods of temporal compositing are required to reveal the full spatial extent of fishing grounds, anchorages, and transit lighting structures. Where vessels are transitory, lighting structures emerge through the accumulation of detections over multiple years.

The multiyear offshore lighting products were initially developed to identify fishing grounds for labelling VBD
detections from fishing boats. The development effort was our response to Philippine and Indonesia fishery agency staff who asked how to distinguish lit fishing boats from other sources of offshore lighting. They noticed that there are numerous VBD detections in locations not known to be fishing grounds. To answer the question, we accumulated VBD detections across Southeast Asia over extended periods of time and found that the fishing grounds emerged as loose clusters having numerous VBD detections. In shallow waters fishing grounds often have locations having many more
detections than their surroundings, referred to as recurring detection locations. These may be lift-net platforms where light is used to aggregate catch. In the Gulf of Thailand these recurring detection locations are arranged in regular grids. In Malaysia the recurring detections are often arranged in regularly spaced lines. This mapping makes it possible to distinguish light detections from fishing boats from other sources.

Based on the features found in the regional prototype, we embarked on an effort to assemble a comprehensive
global mapping of offshore lighting. VBD is quite good at detecting isolated sources of lighting, but does poorly in the detection of glow along shorelines and bright offshore sources. To make the product comprehensive, we combined VBD with the offshore portions of EOG's VIIRS nighttime lights (VNL) product. VNL is designed for annual mapping of onshore lighting from human settlements and includes the glow along shorelines and bright offshore sources.

Taking this approach, we have produced a comprehensive mapping of offshore lighting structures using a
combination of two global VIIRS multiyear nighttime light products – VBD and VNLLarge numbers of distinct features are evident, which werefer to as as lighting structures because they have defined shapes and locations. Overall, VBD has vastly more identifiable lighting structures but is depauperate in glow. The two products are merged at the grid cell level by accepting the product having the higher average radiance. We provide open access to the VBD and VNL products separately and also the combined product so that users can use the one they prefer.

To increase the value of the offshore lighting product, we plan to generate bounding vectors and temporal profiles for many more offshore lighting structures and give each structure a source type (e.g. fishing ground, anchorage or flare) plus a unique identification number. This will make it possible to label the individual VBD detections in the

nightly VBD product with their type associated feature number. We also plan to derive temporal profiles for as many of the lighting structures as possible, to periodically update the temporal profiles and provide access via a web map

service. We will also work to improve the quality of the multiyear offshore lighting product by fully reprocessing the source data with a new cloud detection algorithm and an atmospheric correction.

The current product set provides the most comprehensive mapping to date of offshore light pollution. We expect the data will be useful in marine spatial planning, improved delineation and analysis of fishing grounds, and supply chain issues embedded in the anchorage temporal profiles.


**Author Contributions:** Conceptualization, writing and figures C.D.E.; data preparation T.G. and N.C.; VBD algorithm M.Z.; manuscript review and revision P.S. and M.B.

**Funding:** From 2015 to 2018, the NOAA Joint Polar Satellite Program (JPSS) proving ground program and the U.S.
Agency for International Development's Indonesia and Philippines offices sponsored the development of the VIIRS boat detection algorithm.

**Data Availability Statement:** The multiyear VBD, VNL, and merged grid sets are available via Elvidge *et al.*, 2024 (https://doi.org/10.25676/11124/179157).


**Acknowledgments:** This study made possible by VIIRS data collections made by the NASA/NOAA Joint Polar Satellite System (JPSS).

**Conflicts of Interest:** The authors declare no conflicts of interest.

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
