# Peer review of "A Comprehensive Global Mapping of Offshore Lighting"

_Earth System Science Data, 2024_

## Author Response (AR1)

Offshore Lighting Paper Reviews:

Reviewer 1:

The present work, "A Comprehensive Global Mapping of Offshore Lighting" by Elvidge et al, describes a new product resulting from the merger of two widely used datasets, the VIIRS boat detection (VBD) rasters and the VIIRS cloud-free nighttime lights (VNL).

This product will be of interest for researchers of various fields, in particular those focused on the marine and coastal environment. The description of the merger and filtering criteria, the meaning of the resulting datasets, some interpretation cues, and known artifacts, are sufficiently detailed. Researchers already acquainted with the use of VIIRS-DNB remote sensing products will get the most out of it, but it is also a useful starting point for beginners.

In my opinion this paper can be published essentially in its present form.

There are two minor points the authors may want to comment, for the benefit of readers:

- 2.6 National Offshore Lighting Index: "The percentage of offshore areas having lighting detected from 2012-2022 was calculated for each country by dividing the lit surface area by the total offshore area calculated for that country's exclusive economic zone (EEZ)." -> Does this index include scattered light (typical from shoreline VNL) or only proper, sea-level primary light sources (some VNL + VBD)?

Inserted text: ", from the merged VBD and VNL grid,"

- Fig 7. Koh Tao -> This highly regular square pattern is intriguing. Is there any known reason why it does appear?

Text added to address this point: "Speculatively, this grid pattern may have been developed to deconflict fishing ground utilization, self organized by large fishing companies? The precise origin of the gird remains "to-be-determined". "

Typos and minor issues

- section 2.1 There are two repeated sets of sentences (lines 103 vs 110, and 122-123 vs 109-110).

Removed the second sentence.

- line 198 "Where both VBD and VNL are present, VBD's average radiance is greater than the VNL radiance for 98% of the area. The total area in the merged product covers 1.4% of the offshore total." --> It would be worth noting that VBD is direct or sea-reflected radiance whereas, in terms of extent, VNL offshore is frequently light scattered in the atmosphere.

Inserted sentence "Note that VBD detections are primarily point sources of lighting detected on individual nights, while VNL a cloud-free average that incorporates DNB radiances when no lighting is present."

- line 216-217 -> recheck (duplicate acronym?)

Corrected. The second "VNL" changed to "VBD".

- lines 231-232: [Fig 5] "In the case of shorelines with glow, such as along the shorelines of (…) VBD was selected over VNL. --> it looks like it is the opposite, in Fig5 (?)

Yes, corrected. Extensively edited the paragraph to: "Figure 5 is a color- composite that reveals the source of the VBD / VNL merged product. The merged product reports the average radiance for the product having the larger average DNB radiance. The average radiance coming from VNL is red, and the average radiance from VBD is cyan.  In the case of shorelines with glow, such as along the shorelines of Qatar, Bahrain, and the United Arab Emirates, VNLVBD was selected over VBDVNL. This situation arises based on the specific characteristics of the glow and the spike detection nature of the VBD algorithm.  VNL is selected in the gaps between sporadic VBD detections in the glow surrounding flares. Note that EOG maintains a global catalog of gas flaring sites based on the infrared VIIRS Nightfire product (Elvidge et al., 2015)."

- Figure 16. "VBD radiance temporal profile for a natural gas flare in the Bohai Sea. The radiance starts from 2500 to 3000 from 2012 to mid-2015 (…)" --> it would be useful for readers making explicit which are the radiance units corresponding to these quantities.

Added the radiance units to the Figure 116 caption: "Watts/m$^2$/sr/um".

Reviewer 2:

The paper "A Comprehensive Global Mapping of Offshore Lighting" by Elvidge et al. presents a study on global offshore lighting structures using multiyear data from the VIIRS day/night band (DNB). The authors merge VIIRS boat detection (VBD) data with cloud-free nighttime lights (VNL), allowing for the identification of diverse lighting structures, including fishing grounds, platforms, gas flares, and transit routes.

The dataset described in the paper is poised to become a valuable resource across multiple fields, including studies on the impact of offshore lighting on light pollution and its consequent effects on coastal ecosystems.

Overall, this paper is an excellent contribution to the field, providing an effective methodology for mapping offshore lighting. I recommend its publication, with a few minor changes/suggestions:

* Section 2.1: The first and second paragraphs are almost identical, with several sentences repeated across both. Additionally, the third paragraph reiterates information from earlier in the section, such as the reference to Table 1. For clarity, please revise this section to reduce redundancy.

Thank you for spotting that. Revise to reduce redundancy.

* Section 3.3.5: This subsection is very brief, with just one sentence providing limited information not already shown in Figure 9. To maintain consistency with other subsections, expanding Section 3.3.5 to be more descriptive would improve the overall clarity and cohesiveness of the paper.

Added: "Similar convoluted patterns can be found, but the example in Figure 9 is exemplary.  The origin of the convoluted patterns may trace back to bathymetry, ocean currents,  or other not- yet identified reasons."